# Paper-based electrochemical device for early detection of integrin αvβ6 expressing tumors
Stefano Cinti [1,7] ✉, Stefano Tomassi [1,7], Chiara Ciardiello[2,7], Rossella Migliorino[2], Marinella Pirozzi [3], Alessandra Leone[2], Elena Di Gennaro[2], Virginia Campani[1], Giuseppe De Rosa[1], Vincenzo Maria D'Amore[1], Salvatore Di Maro[4], Greta Donati[1], Sima Singh[1], Ada Raucci[1], Francesco Saverio Di Leva [1], Horst Kessler [5], Alfredo Budillon[6] & Luciana Marinelli [1] ✉

Despite progress in the prevention and diagnosis of cancer, current technologies for tumor detection present several limitations including invasiveness, toxicity, inaccuracy, lengthy testing duration and high cost. Therefore, innovative diagnostic techniques that integrate knowledge from biology, oncology, medicinal and analytical chemistry are now quickly emerging in the attempt to address these issues. Following this approach, here we developed a paper-based electrochemical device for detecting cancer-derived Small Extracellular Vesicles (S-EVs) in fluids. S-EVs were obtained from cancer cell lines known to express, at a different level, the αvβ6 integrin receptor, a well-established hallmark of numerous epithelial cancer types. The resulting biosensor turned out to recognize αvβ6-containing S-EVs down to a limit of $0.7*10^3$ S-EVs/mL with a linear range up to $10^5$ S-EVs /mL, and a relative standard deviation of 11%, thus it may represent a novel opportunity for αvβ6 expressing cancers detection.

The detection of cancer in its early stages can greatly increase the chance of survival and decrease the medical management costs. Nowadays, there are numerous methods available for cancer detection, including radiography, echography, computed tomography scan (CT), magnetic resonance imaging (MRI), and positron emission tomography (PET). However, these techniques cannot be always employed due to patient-specific factors like comorbidity which can hinder the examination process. Additionally, some methods cannot be regularly utilized as screening techniques due to invasiveness or high cost (e.g., PET scan), while others are ineffective in detecting cancer at its early stages. Therefore, novel, non-invasive, rapid, effective, low-cost techniques for the quantitative detection of specific tumor markers are urgently needed to ensure early cancer diagnosis and/or to monitor disease progression and response to treatment[1]. In this perspective, various biochemical methods based on the detection and quantification of biomarkers, substances present in abnormal amounts in people with cancer or a precancerous condition, are currently being studied and developed. Most

common technologies for the detection of biomarkers include gel electrophoresis[2], surface plasmon resonance[3], colorimetric assays[4], electrochemical assays[5], mass-based/piezoelectric, electrochemical, or optical approaches[6–8], and enzyme-linked immunosorbent assay (ELISA)[9]. In addition, biosensors, analytical devices used for the detection of any sort of biological interaction, are increasingly emerging as an attractive alternative to such technologies[1]. Notably, biosensors can be portable, thus facilitating the "self-use" and, in turn, mass cancer screening[10,11], especially through liquid biopsy. In particular, the exploitation of novel paper-based platforms is capable to provide further advantages, i.e., affordability, sustainability, easiness to use, to the biosensor field, as widely reported in literature[12–14].

The efficacy of identifying small molecules, proteins, nucleic acids, cells, etc. related to liquid biopsy is strictly dependent on both the amount of the target and the transduction that is used to develop the paper-based assay. In fact, while colorimetric paper-based biosensors offer advantages of simplicity and straightforward interpretation, they are mostly characterized

[1]Dipartimento di Farmacia, Università degli Studi di Napoli "Federico II", Via D. Montesano 49, 80131 Naples, Italy. [2]Experimental Pharmacology Unit, Istituto Nazionale Tumori–IRCCS– Fondazione G. Pascale, Via Mariano Semmola, 53, 80131 Naples, Italy. [3]Second Unit, Institute of Experimenal Endocrinology and Oncology "G. Salvatore" (IEOS), National Research Council (CNR), Naples, Italy. [4]Department DiSTABiF, University of Campania "Luigi Vanvitelli", Via Vivaldi 43, 81100 Caserta, Italy. [5]Institute for Advanced Study and Center of Integrated Protein Science, Department Chemie, Technical University of Munich, Lichtenbergstraße 4, 85748 Garching, Germany. [6]Istituto Nazionale Tumori –IRCCS– Fondazione G. Pascale, Via Mariano Semmola, 53, 80131 Naples, Italy. [7]These authors contributed equally: Stefano Cinti, Stefano Tomassi, Chiara Ciardiello. ✉e-mail: stefano.cinti@unina.it; luciana.marinelli@unina.it

with semi-quantitative outcomes, i.e., lateral flow assays[15]. Conversely, electrochemical methods are usually capable of providing higher sensitivity and are prone to quantitative and semi-quantitative determination of analytes, as in the case of nucleic acid detection in biological samples down to pM-nM[16,17].

In parallel, the identification of novel cancer-specific biomarkers represents a relevant task as many of those currently available suffer of poor stability and low concentration in biological fluids[18]. In this respect, the detection of a specific biomarker in cancer-derived Extracellular Vesicles (EVs) could offer numerous advantages especially in terms of stability. Indeed, EVs are secreted by various cells but abundantly by cancer cells and can be classified into diverse categories based on their biogenesis and size[19,20]. Medium/large EVs (>200 nm) such as microvesicles, apoptotic bodies and large oncosomes are generally both assembled at and released from the plasma membrane (PM)[20], while EVs smaller than 200 nm (S-EVs) are mainly represented by exosomes generated through exocytosis of vesicular bodies[21]. The role of cancer-derived EVs, especially S-EVs, in preparing the pre-metastatic niche and carrying information regarding the tumor microenvironment has been recently established[22]. Thus, their detection could provide important information not solely on cancer diagnosis but also on disease aggressiveness and therapy efficacy, thus allowing to consider S-EVs as next-generation cancer biomarkers. At this regard, in 2015, Lyden and collaborators, taking human breast cancer as a model, demonstrated that the metastatic sites reached by S-EVs are determined by their integrin expression pattern and that S-EVs are typically enriched in extracellular matrix binding proteins and cell adhesion receptors such as integrins[22], which are thus eligible as ideal biomarkers for cancer detection. In this regard, the biosensor field is making big strides in the detection of EVs in biological matrices using nanosized materials and biomimetic recognition probes, i.e., aptamers[23–25].

Among the various integrins, we recently focused on αvβ6, an epithelium-specific receptor, which is expressed at high levels in developing lung, skin and kidney, while its expression is negligible in healthy adult tissues[26]. Interestingly, αvβ6 integrin is highly upregulated in a number of epithelial carcinomas (i.e., breast, prostate, lung, oral, skin, colon, and stomach) where it is associated with poor prognosis[27]. Herein, a multi-disciplinary approach, encompassing biology, oncology, medicinal and analytical chemistry, was adopted to develop a nano-engineered paper-based electrochemical device, which can detect αvβ6-containing cancer-derived S-EVs.

## Results and discussion

Our biosensor was manufactured starting from a highly affine and αvβ6-selective cyclic ligand, [RGD-Chg-E]-CONH$_2$ (1) (IC$_{50}$s of 1.3 nM for αvβ6, 364 nM for αvβ3, 105 nM for α5β1, and 174 nM for αvβ8, respectively), recently developed by some of us[28]. This peptide was specifically modified and anchored onto a paper-based printed chip to detect, in a rapid manner, αvβ6-containing S-EVs (αvβ6 S-EVs) in a few microliters sample. For this

purpose, 1 was functionalized with linkers of variable length ranging from 13 to 28 atoms (2-5) and bearing a terminal thiol group to covalently bind a layer of gold nanoparticles at the printed chip (Fig. 1). Since long aliphatic chains may cause poor water solubility and potential intermolecular hydrophobic interactions between the probes, especially in aqueous medium, polyethylene glycol(PEG)-based chains were used as suitable spacers. The desired compounds were synthesized on a solid support through an ultrasound-assisted solid-phase peptide synthesis (US-SPPS) protocol[29]. The pharmacophoric sequence (RGD-Chg-E) was assembled on a lysine side chain, used as a common spacer, and Nα-protected with an *ortho*-nitrobenzenesulfonyl-(*o*NBS-) protective group. Upon allyl removal at the Glu sidechain and cyclization step, the *o*NBS deprotection allowed for the C-terminal elongation with varying linkers prior to the thiol group functionalization and final reductive cleavage from the resin (Fig. 1 and Scheme S1)[30].

The newly synthesized probes 2-5 were then engineered, one at a time, onto a printed electrochemical device (Fig. 2). Briefly, the printed electrochemical device has been obtained by screen-printing carbon and silver/silver chloride based conductive inks onto office-paper substrates, by using two ad-hoc realized masks, and the cost of the electrode has been calculated to be of ca. 0.02 Euro/each[31]. To define the testing area, a wax-based ink has been printed using a solid-ink printer. The home-made produced screen-printed electrodes were characterized by an area of the working electrode equal to ca. 12 mm$^2$, and the complete manufacturing procedure is reported in the technical section below. Then, electrochemical impedance spectroscopy (EIS) experiments were performed in the presence of an external redox couple, namely potassium ferrocyanide and potassium ferricyanide (ferro/ferricyanide). Briefly, when αvβ6 is recognized by the immobilized probe onto the device, the diffusion of ferro/ferricyanide towards the electrode area is hindered, resulting in an increased impedance signal (i.e., the diameter of the semi-circle) with respect to the signal obtained in absence of the protein, as shown in Fig. 2b.

Initially, compounds 2-5 were tested at 100 nM concentration to define the optimal distance (i.e., the ideal linker length) between the integrin-recognizing probe and the layer of gold nanoparticles at the printed chip. Then, impedance experiments were carried out in presence of a fixed concentration of αvβ6 equal to 5 ng/mL. All the assays were performed in 50 mM phosphate buffer, and 150 mM sodium chloride (pH = 7.4). As shown in Fig. 3a, the strongest signal was recorded when the probe 3 (17-carbon length linker) was engineered onto the electrode surface. This result was somehow expected because too short linkers could hamper the diffusion of redox compounds towards the electrode, while too long ones commonly adopt undesired folding. In our case, the charge transfer resistance (Rct) was determined using ferro/ferricyanide couple as redox system, in the presence of integrin concentration ranging from 0.01 to 50 ng/mL. The increase in the target concentration led to an augmented width of the spectra (the diameter of semi-circle, defined as the Rct), as reported in Fig. 3b. Once the optimal probe was identified (3), a calibration curve was obtained in the presence of a

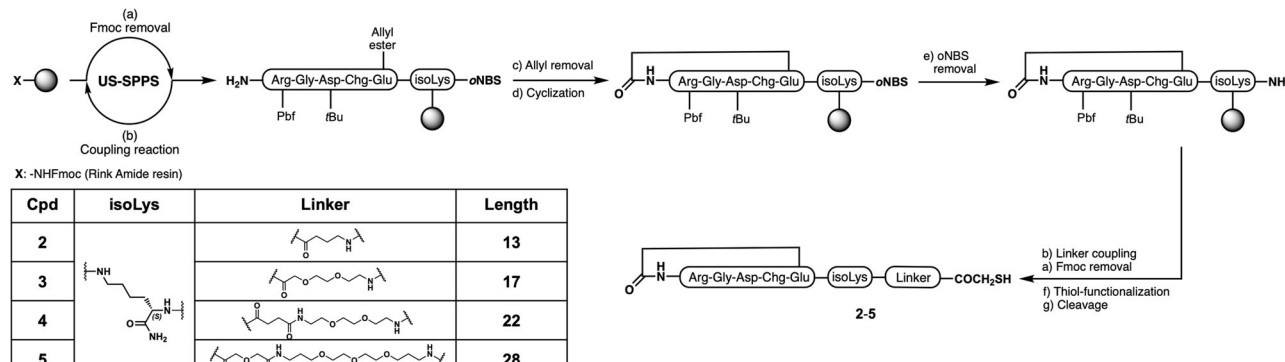

**Fig. 1 | Synthesis of the thiol-containing probes 2-5.** Schematic representation of the ultrasound-assisted solid-phase peptide synthesis (US-SPPS) protocol used to produce peptides 2-5.

wide range of αvβ6 concentration (Fig. 3c). Spectra were fitted using the equivalent electrical circuit showed in the inset of Fig. 3a which comprises a) the electrolyte resistance (Rs) in series with a parallel combination of the charge transfer resistance (Rct), b) the diffusion of the analytes in solution, corresponding to Warburg impedance (Zw), and c) a constant phase element (CPE). The impedance spectra obtained are composed of a semicircle at high frequencies, followed by a 45° straight line at low frequencies. The equivalent circuit that describes this trend is a typical Randles circuit, composed by the resistance of the solution Rs in series with the capacitance of the electric double layer CPE and the charge transfer resistance Rct (in series with an open Warburg element Zw, that describes the mass transport), respectively in parallel. The double-layer electrical capacitance was fitted with a CPE, since the electrode surface is not completely homogeneous. Rct

is the only parameter proportional to the analyte concentration and was used to produce the calibration curves, while the other parameters remain constant, as expected. With this experimental setup, a semi-logarithm correlation was described within the 1–20 ng/mL concentration range by the following equation y = 10944 x + 3940 (R² = 0.986), where y represents the Rct and x represents log[αvβ6].

The apparent binding constant and detection limit were equal to 5.3 ± 0.6 and 1.0 ± 0.1 ng/mL, respectively. It should be clarified that, being the probes immobilized onto a surface, both the detection limit and specificity of these sensors usually are not dependent on the "true" probe-target affinity but depend on probe concentration; consequently, the response of the sensor platform depends on the density with which the probes are packed on the surface of the sensor, thus leading to an estimation of an "apparent" or "observed" affinity, as previously reported[32]. The detection limit was approximated to the integrin concentration corresponding to the Rct related to the mean of three analyzed blank samples plus three times its standard deviation. The repeatability of the platform was also evaluated by measuring 20 ng/mL of αvβ6 integrin with 8 different electrodes: a Rct of 39.5 ± 4 kΩ has been obtained, corresponding to a relative standard deviation (RSD) of ca.11%, thus confirming the robustness of the paper-based chip manufacture. Subsequently, the device was evaluated for the capability of selectively recognizing αvβ6 with respect to other RGD integrins subtypes. In fact, although substantial changes in the integrin binding profile of **3** with respect to its parent compound **1**[28] (the two molecules only differ for the presence or absence of a 17-carbon length linker) were not expected, this was again evaluated when **3** was directly engineered onto the electrode surface.

As shown in Fig. 4, the device was able to specifically detect αvβ6 in presence of interfering species. Finally, the device was assessed for the detection of αvβ6-containing s-EVs by applying the optimized setup, in terms of spherical gold nanoparticles, density and length of the probe. Particularly, it was evaluated in presence of S-EVs derived from in vitro cell cultures: (a) αvβ6 expressing cancer cell lines (PC3 and DU145R80 herein named R80)[33–35], (b) αvβ6 very low expressing (negligible) cancer cell lines (A549, HCT116, H460) (c) αvβ6 very low expressing (negligible) non-cancer cell line (HEK293) derived from human embryo's kidney (d) liposomes of lipidic composition and size similar to those of S-EVs (totally negative control)[36,37].

Firstly, we confirmed the level of αvβ6 integrin expression in the panel of chosen cells (Fig. S2, Panel A)[37,38]. Then, S-EVs from cancer/non cancer

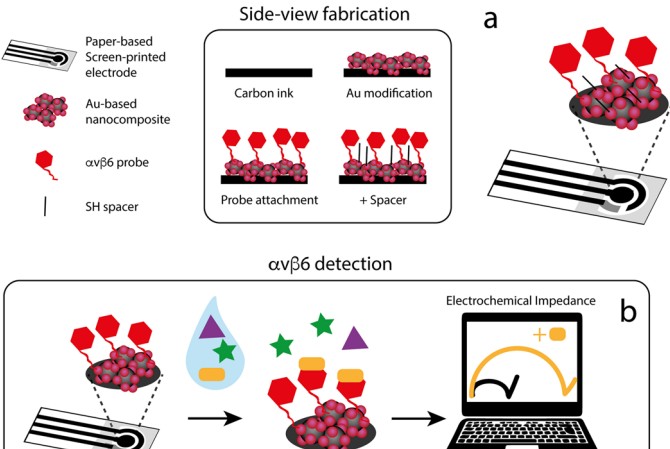

**Fig. 2 | Schematic representation of the platform. a** Engineering of the paper-based electrochemical strip following the screen-printing of conductive ink, modification with a dispersion of gold nanoparticles, covalent engineering of the αvβ6-selective probe and saturation of the surface with mercaptohexanol (6-MCH);
**b** Impedimetric measurements show an increase of the semi-circle (charge transfer resistance, Rct) when the binding between probe and protein occurs (orange line) in comparison with absence of target (black line).

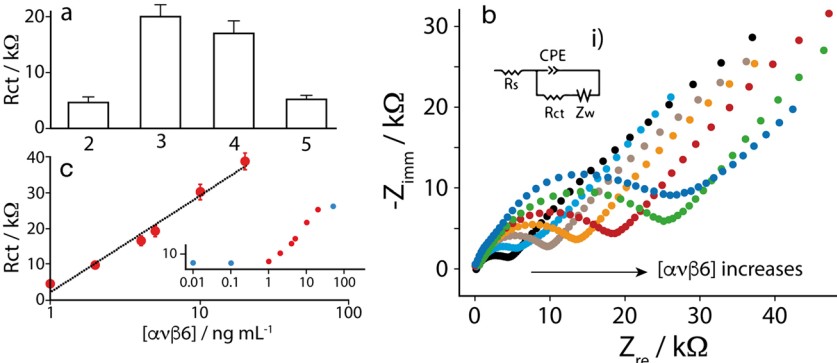

**Fig. 3 | Analytical performance in presence of αvβ6. a** Optimization of the probe length to be immobilized onto the strip. For each experiment each probe at 100 nM concentration has been immobilized onto the strip; the impedimetric measurements have been carried out in presence of 5 ng/mL αvβ6. **b** Nyquist plots for the nanoengineered biosensor **3** challenged in absence (black dots) and in presence of different concentration of αvβ6, i.e., 1 (cyan dots), 2 (gray dots), 4 (orange dots), 5 (red dots), 10 (green dots) and 20 ng/mL (blue dots). The measurements were carried out as follows: the nanoengineered biosensor was covered with a solution containing the chosen level of αvβ6. After 30 min at room temperature, the biosensors were washed with phosphate buffer, and then covered with a solution

containing 1 mM [Fe(CN)₆]³⁻/⁴⁻ dissolved in 0.1 M KCl. A potential of 0.2 V and a AC amplitude of 10 mV in a frequency range of 100 kHz to 0.1 Hz have been utilized. The X and Y axes represent, respectively, the real and imaginary components of impedance. The inset i shows the Randles equivalent electrical circuit that has been used to fit the spectra, comprising the electrolyte resistance, Re, in series with a parallel combination of Rct (charge transfer resistance), Zw (diffusion of the analytes in solution and corresponding to Warburg impedance straight line of the curves) and CPE (Constant Phase Element). **c** Linear range comprised between 1 and 20 ng/mL αvβ6. Inset shows the complete semi-log correlation in a wider range of αvβ6 concentrations, namely 0.01–50 ng/mL.

cells were collected by an isolation kit (see Methods section for details) and characterized (Fig. 5). The size of S-EVs was measured by Tunable Resistive Pulse Sensing (TRPS, Fig. 5a), displaying that it mostly ranged from 60 to 200 nm. In fact, although EVs >200 nm were also detected, their concentration was not relevant in our preparations. Transmission electron microscopy (TEM) of collected S-EVs confirmed TRPS size evaluation and that our extracts were enriched in intact S-EVs of 100 nm size (Fig. 5b, upper panels). Notably, when tested by Immuno-electron microscopy (IEM), the majority of S-EVs derived from very low expressing (negligible) HEK293 and H460 cell lines were positive to αvβ6 integrin (Fig. 5b, lower panels), as demonstrated by black gold labels (similar results were obtained for both A549- and HCT116-derived S-EVs, data not shown). None gold labels were instead found on the carbon grids of negative controls (Figure S2, Panel B), indicating very low background. The presence of αvβ6 in S-EVs originated from very low expressing (negligible) cancer cells is in line with the observation that S-EVs are typically enriched in integrins with respect to the parent cell membrane[39] as they must guarantee the S-EVs-matrix adhesion to drive organotropic metastasis[22]. Considering that: (i) cancer cells release S-EV to a larger extent than healthy cells, (ii) S-EV are notoriously enriched in integrins, and (iii) αvβ6 expression is negligible in healthy adult tissues, while it is highly upregulated in many epithelial carcinomas, αvβ6 on S-EVs may represent a novel, precious cancer biomarker, for which a quantitative threshold must be defined.

In fact, medium αvβ6-expressing R80- and PC3 cancer cells produced S-EVs featuring higher amount of αvβ6 integrin compared to S-EVs derived from low αvβ6 expressing cancers such as A549, HCT116, HEK293 and H460 cell lines (Fig. 5c, d). These data are particularly significant as they illustrate that, despite the general enrichment of integrins in S-EVs, the quantity of αvβ6 on their surface consistently mirrors that of the parent cells. As expected, all S-EVs preparations were positive to both TSG101 and CD81

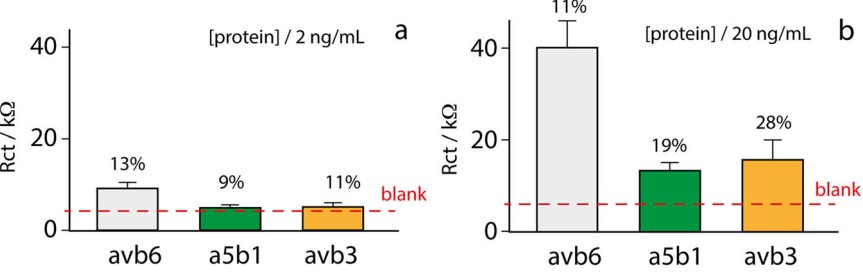

**Fig. 4 | Selectivity study of the optimized platform.** **a** In presence of 2 ng/mL and (**b**) in presence of 20 ng/mL of αvβ6 (gray bars), α5β1 (green bars) and αvβ3 (orange bars) integrins. The bars are the results of 5 replicates. The dashed red lines indicate the signal observed in absence of targets. The concentrations of integrins have been selected in agreement with the linearity range of the platform.

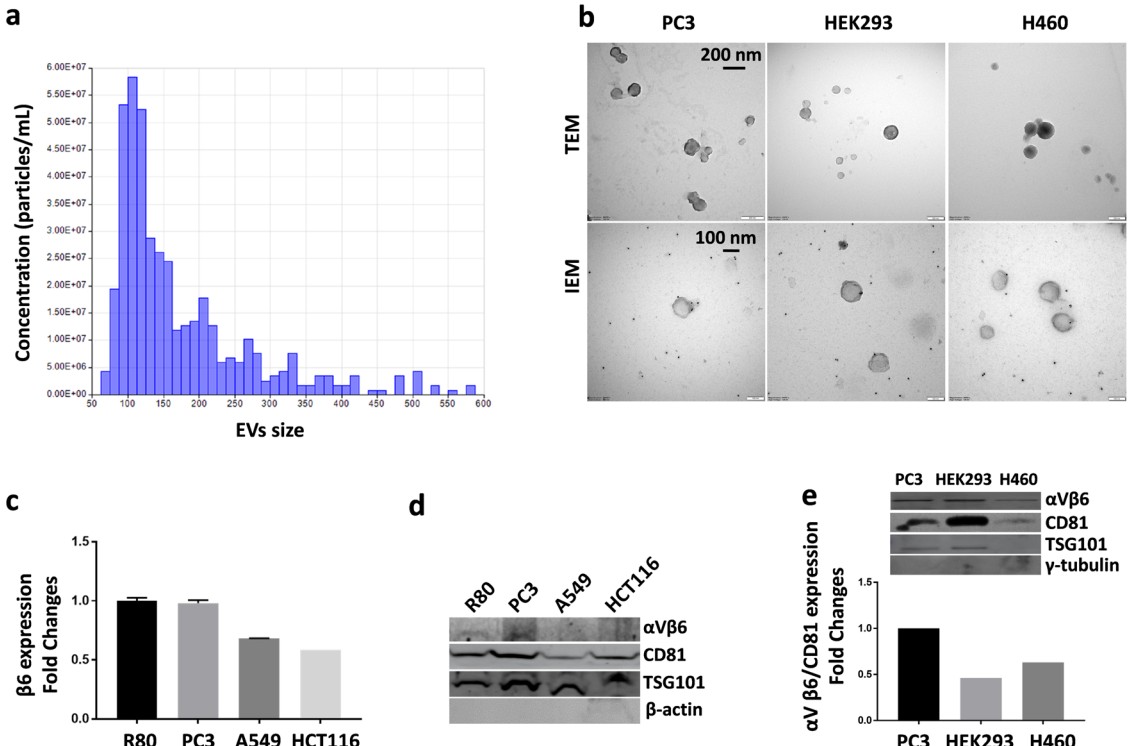

**Fig. 5 | S-EVs characterization.** **a** Representative image of TRPS measurements showing particle size and concentration of S-EVs derived from DU145R80 (R80) cancer cell line (see text). **b** Upper panels: representative images of Transmission Electron micrograph (TEM) of S-EVs isolated from cell culture media. Magnification 49000X, scale bar 200 nm. The morphology is observed by negative staining. Lower panels: representative images of Immuno Electron micrograph (IEM) of S-EVs, labeled using antibodies specific to αvβ6 integrin and antibody binding was confirmed by Protein A gold-conjugate to 10 nm gold particles. Magnification 120000X, scale bars 100 nm. **c** Integrin β6 expression by ELISA expressed in terms of fold changes of S-EVs from PC3, A549 and HCT116 compared to R80-derived S-EVs. **d** Western blot analysis of S-EVs lysates from R80, PC3, HCT116 and A549 and tested with antibodies as indicated in the figure. Both TSG101 and CD81 were tested as positive EVs markers. β actin served as negative control. **e** Densitometry of Integrin αVβ6/CD81 expression in EVs, expressed as fold changes in either HEK293- or H460- S-EVs compared to PC3-derived S-EVs. Inset shows Western blot analysis of S-EVs lysates from PC3, HEK293 and H460 tested with antibodies as indicated in the figure. Both TSG101 and CD81 were tested as positive EVs markers. ɣ-tubulin served as negative control .

**Fig. 6 | Analytical performance in presence of S-EVs. a** Nyquist plots for the nanoengineered biosensors challenged in absence (black dots) and in presence of different concentration of R80 S-EVs i.e., $10^3$ (cyan dots), $10^4$ (orange dots), $10^5$ (blue dots) and $10^6$ (red dots). The measurements were carried out as described in caption of Fig. 2a. Inset: it shows the schematic of measurements at the paper-based chip. **b** Semi-log correlation in the range of R80 S-EVs comprised between 10 and $10^7$ S-EVs/mL; **c** Histograms relative to the response in presence of $10^4$ S-EVs/mL belonging to different cell-lines including R80, PC3, A549, HCT116, H460 and HEK293. Liposomes were also tested as negative controls.

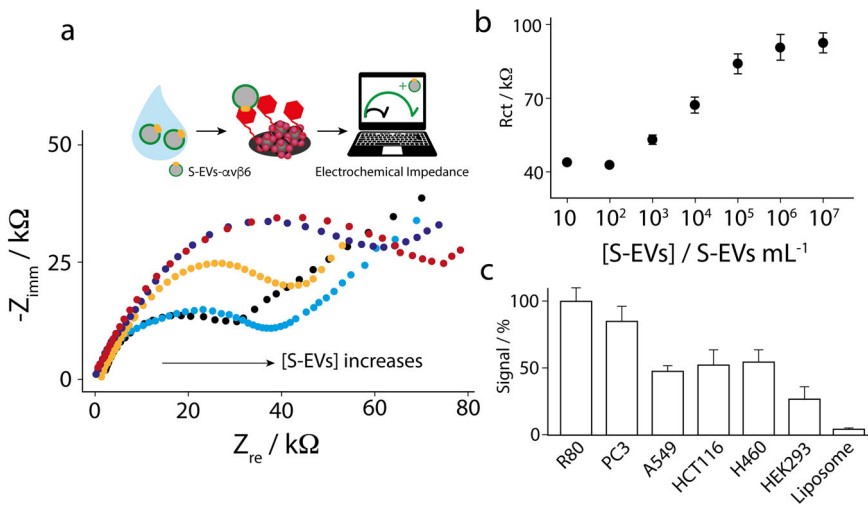

EVs markers (Fig. 5d, e). Both β-actin/γ tubulin signals served as negative controls for S-EVs preparations (Fig. 5d, e). Benefiting of R80, PC3, A549, HCT116, H460 and HEK293-derived S-EVs, the device was evaluated for the determination of αvβ6-S-EVs by applying the already described optimized setup.

As reported in Fig. 6, Rct increases with S-EVs concentration. It is also interesting to note that its values are higher than those recorded in presence of the integrin alone. This can be ascribed to the bigger size of S-EVs (compared to the isolated protein), which hinders the redox probe from reaching the surface of the electrode. A good semi-logarithm correlation was obtained by measuring the S-EVs from R80 cells; particularly, the equation $y = 5860\ x + 11400$ ($R^2 = 0.985$) was defined in the concentration range comprised between $10^2$ and $10^5$ S-EVs/mL (Fig. 6b). The detection limit (LOD) was equal to $0.7*10^3$ S-EVs/mL, calculated as 3.3-times the standard deviation of the blank solution divided by the slope of the calibration curve, while the quantification limit (LOQ) was equal to $2.2*10^3$ S-EVs /mL, according to the following relation LOQ = 3*LOD. To evaluate the platform performance towards different αvβ6-containing S-EVs, all the experiments were carried out by utilizing the optimized experimental setup, whereas the concentration of each S-EVs line was fixed to $10^4$ S-EVs/mL, in accordance with the linear range of response ($10^2$ and $10^5$ S-EVs/mL, Fig. 6b). As displayed in Fig. 6c, the paper-based strip was able to recognize αvβ6-medium-expressing S-EVs (from R80, PC3 cells), from αvβ6-low-expressing S-EVs (from A549, HCT116, H460 and HEK293 cells). Also, it was capable to distinguish samples with low content of αvβ6, from totally negative control (liposomes). As far as we know, the development of a portable, electrochemical device for the recognition of αvβ6 appears as novel in literature. As shown in Table S2 (see SI), the detection limit obtained in this work satisfactorily compares with other sensing approaches applied towards integrin receptors, and the electrochemical methods reported here is more user friendly of the largely reported SERS and FRET-based architectures.

In conclusion, merging different disciplines, namely synthetic and analytical chemistry, cancer biology and bioengineering, we developed a generalizable multi-disciplinary approach for liquid-biopsy on printed strip. A library of αvβ6 selective ligands, featuring SH-linkers of different length suitable for biosensors engineering, was designed and synthetized. These compounds were fixed onto our electrochemical device to assess their capability to properly recognize αvβ6-S-EVs in solution. Cycles of optimization in terms of gold nanoparticles, and probe density were performed, leading to the selection of **3** as the best candidate for detection studies. This biosensor showed detection and quantification limits equal to 0.7 and $2.2*10^3$ S-EVs /mL, respectively, while the linear range was comprised between $10^2$ and $10^5$ S-EVs/mL. The next step in our research pipeline will

necessarily regard the testing of the device on human biofluids (e.g., urine in case of PCa screening) from αvβ6-cancer bearing patients and healthy donors and will be object of a dedicated study. This step will allow us to determine a clinically validated threshold for αvβ6-detection to discriminate between the two source populations.

## Methods
### Materials
Standard Nα-Fmoc-protected amino acids, $O$-benzotriazole-$N,N,N',N'$-tetra-methyl-uroniumhexafluorophosphate (HBTU, purity 99%), $N,N$-diisopropylethylamine (DIEA, purity 99%), trifluoroacetic acid (TFA, purity 99%), piperidine, Fmoc-L-Arg(Pbf)-OH, Fmoc-Gly-OH, Fmoc-L-Asp(O$t$Bu)-OH, Fmoc-L-Glu(OAll)-OH (CAS # 133464-46-7), Fmoc-L-Chg-OH (CAS # 161321-36-4), Fmoc-O2Oc-OH (CAS # 166108-71-0), Fmoc-Ebes-OH (CAS # 613245-91-3) and Fmoc-GABA-OH (CAS # 166108-71-0) were purchased from IRIS Biotech (Marktredwitz, Germany).

Fmoc-Rink amide-Am resin, Fmoc-NH-(Peg)₂-COOH (20 atoms) (CAS # 916585-44-9), triisopropylsilane (TIS) (purity 98%), 1-hydroxybenzotriazole hydrate (HOBt) (purity > 97% dry weight, water ≈ 12%), (1$H$-7-Azabenzotriazol-1-yl-oxy)tris-pyrrolidinophosphonium hexafluorophosphate (PyAOP) (purity 96%), 1-hydroxy-7-azabenzotriazole (HOAt, purity 96%), tetrakis(triphenylphosphine)palladium(0) (purity 99%), dimethylbarbituric acid (DMBA, purity 99%, water content < 6%), thiophenol (purity 97%), potassium carbonate ($K_2CO_3$) (purity 99.995%, trace metal basis), DL-Dithiothreitol solution (1 M in water) (CAS # 3483-12-3), anhydrous $N,N$-dimethylformamide (DMF), and anhydrous dichloromethane (DCM) were purchased from Sigma-Aldrich (Milano, Italy). Soy phosphatidylcholine was purchased from Lipoid GmbH (Germany). $o$NBS-L-Lys(Fmoc)-OH was synthesized according to a protocol reported elsewhere[30].

Peptide synthesis solvents, water and acetonitrile for HPLC, were reagent grade and were acquired from commercial sources (Honeywell International Inc.) and used without any further purification unless otherwise stated. Peptides were assembled on Macherey-Nagel Chromab. vacuum manifold and using an ultrasonic bath *SONOREX* RK 52 H (interior dimensions 150 × 140 × 100 mm and operating volume 1.2 L) by BANDELIN electronic (Germany). Peptides were purified by preparative HPLC (Shimadzu HPLC system) equipped with a C18-bounded preparative RP-HPLC column (Phenomenex Kinetex 21.2 × 150 mm, 5 μm). Peptides were analyzed by analytical HPLC (Shimadzu Prominance HPLC system) equipped with a C18-bounded analytical RP-HPLC column (Phenomenex Kinetex, 4.6 × 150 mm, 5 μM) using a gradient elution (10–90% acetonitrile in water (0.1% TFA) over 20 min; flow rate = 1.0 mL/min; diode array UV detector). Molecular weights of compounds were

confirmed by ESI-MS using an LTQ XL Linear Ion Trap Mass Spectrometer (ThermoFisher).

### Solid phase synthesis of compounds 2-5

A Rink Amide AM-PS resin (521 mg, 0.25 mmol, 0.48 mmol/g) was swelled in a mixture of DCM/DMF 1:1 for 40 min and then washed with DCM (3 × 1 min) and DMF (3 × 1 min). Initial resin-bound Fmoc protecting group was removed by gently shaking the resin in the presence of a 20% (v/v) piperidine solution in DMF (1 × 5 min; 1 × 25 min). The following synthetic procedure was based on an ultrasound-assisted solid phase peptide synthesis protocol, previously described by some of us[29].

Specifically, $o$NBS-L-Lys(Fmoc)-OH (345 mg, 0.625 mmol, 2.5 equiv according to the initial loading of the resin) was coupled to the resin using HBTU (237 mg, 0.625 mmol, 2.5 equiv) and HOBt (96 mg, 0.625 mmol, 2,5 equiv) as coupling reagents, DIPEA (218 µL, 1.25 mmol, 5 equiv) as base and DMF (4 mL) as solvent. The so-obtained mixture was added to the resin and the plastic SPPS reactor irradiated within a SONOREX ultrasonic bath for 10 min before being washed with DMF (3 × 1 min) and DCM (3 × 1 min). Subsequently, Fmoc protective group was removed using a 20% (v/v) piperidine solution in DMF and irradiating the resin with the ultrasounds in the presence of this mixture (2 × 2 min). Linear elongation of peptides was fulfilled following the so-described ultrasound-assisted protocol by iterative cycles of Fmoc deprotections and coupling reactions with Nα-Fmoc amino acids and completion of these steps were qualitatively monitored by Kaiser ninhydrine and/or TNBS test.

After Fmoc deprotection from the last amino acid (arginine), glutamate side chain allyl ester removal was carried out by treating the resin with a solution of Tetrakis(triphenylphosphine)Palladium(0) (29 mg, 0.025 mmol, 0.1 equiv relative to the initial loading of the resin) and DMBA (390 mg, 2.5 mmol, 10 equiv relative to the initial loading of the resin) in dry DCM/DMF (2:1, 4 mL). The mixture was allowed to gently shake under argon atmosphere for one hour and this step repeated one more time. The resin was washed with DMF (3 × 1 min), DCM (3 × 1 min) and then treated twice with a solution 0.06 M of potassium $N,N$-diethyldithiocarbamate in DMF (76 mg in 6.0 mL of solvent) for 30 min to wash away the catalyst traces. Head-to-side-chain cyclization between arginine α-amine and glutamate w-carboxylic acid was carried out using 3 equiv PyAOP and 6 equiv of DIPEA. PyAOP (391 mg, 0.75 mmol, 3 equiv) was dissolved in a mixture DMF/DCM 9:1 (5 mL), then DIPEA (261 µL, 1.5 mmol, 6 equiv) was added, the so-obtained mixture was added to the resin and shaken for 12 h. The resin was washed with DMF (3 × 1 min) and DCM (3 × 1 min) and then shaken in the presence of DMF (3 mL) for 20 min. Completion of the reaction was qualitatively determined by Kaiser test and TNBS test.

The release of α-amino group from the C-terminal lysine residue was obtained removing the $o$NBS protective group by treating the resin with 5.0 mL of a 5% (v/v) solution of thiophenol in dry DMF in the presence of 2 equiv (relative to thiophenol) of $K_2CO_3$. Prior use, the so obtained suspension was sonicated and then centrifuged (6000 rpm × 15 min). This step was repeated three times (10 min of shaking for each treatment) then the resin was washed exhaustively with DMF (5 × 1 min), MeOH (3 × 1 min) and DCM (3 × 1 min). Primary amine release was qualitatively determined by positive Kaiser test.

The resin was exhaustively dried on the vacuum manifold and then placed under reduced pressure until constant weight was achieved. Then the resin was partitioned in four equal weighted aliquots and swelled again in a mixture of DMF/DCM (1:1) for 40 min. Each aliquot underwent the coupling reaction with its pertinent Fmoc-protected linker using the above-mentioned US-SPPS procedure. Specifically, compounds **2-5** were synthesized performing coupling reactions as stated above with corresponding Fmoc-protected linkers (0.125 mmol, 2.5 equiv), subsequent Fmoc-deprotection allowed the release of primary amino group which then underwent a coupling reaction with Trt-SCH₂COOH (42 mg, 0.125 mmol, 2.5 equiv) in the presence of HBTU (47 mg, 0.125 mmol, 2.5 equiv) and DIPEA (109 µL, 0.625 mmol, 5 equiv) in DMF (1.5 mL) overnight. Positive outcome of each synthetic step was qualitatively determined by Kaiser test.

### Resin cleavage of compounds 2-5

The so-obtained resin-bound peptides were washed with DMF (3 × 1 min), DCM (3 × 1 min), and Et₂O (3 × 1 min) and then dried exhaustively. Then crude peptides were cleaved from the solid support and side-chain protective groups removed in one-pot reaction by treatment with a solution of TFA/TIS/Dithiothreitol solution (1 M in water) (95:2.5:2.5; 1.5 mL) for 3 h at room temperature. The resin was filtered, and the crude peptides precipitated from the TFA solution, diluting to 13 mL with cold Et₂O and then centrifuged (6000 rpm × 15 min). The supernatant was carefully removed, and the precipitate suspended again in 13 mL of cold Et₂O as described above. The resulting wet solid was dried for 1 h under reduced pressure, redissolved in water/acetonitrile (9:1) and purified by reverse-phase HPLC (solvent A: water + 0.1% TFA; solvent B: acetonitrile + 0.1% TFA; from 10 to 90% of solvent B over 25 min, flow rate: 10 mL min⁻¹) unless otherwise stated. Fractions of interest were evaporated from organic solvents under reduced pressure, frozen, and then lyophilized. Obtained products were characterized by analytical HPLC (solvent A: water + 0.1% TFA; solvent B: acetonitrile + 0.1% TFA; from 10 to 90% of solvent B over 20 min, flow rate: 1 mL min⁻¹, unless otherwise stated) and ESI mass spectrometry. Yields, purity, retention times and analytical spectra are reported as follows.

A detailed synthetic scheme (Fig. S1) and the analytical data of compounds **2-5** are reported in Supplementary Information.

### Paper-based chip development

**Production of the paper-based strip.** Office paper (Copy 2, 80 g/m², Fabriano, Italy) was selected as the support to screen-print the conductive inks. Initially, office paper was wax-patterned with a wax-printer (Xerox Color-Qube 8580). The paper-based support was patterned with 1 layer of wax. After a curing stage in the oven at 100 °C for 2 min, the wax penetrated the paper-based structures producing hydrophobic areas that contained the testing area. The electrodes were manually screen-printed using Ag/AgCl ink (Electrodag 477 SS, Acheson, Italy) for the reference electrode and the electrical connections, and carbon ink (Electrodag 421, Acheson, Italy) for the working and counter electrodes. The paper-based devices were printed in strips composed by 8 electrodes. Consequently, each paper-based strip was cut to obtain a 25 × 10 mm-device.

**Engineering the Paper-Based strip with the AuNPs/probe hybrid.** A drop containing 8 µL of AuNPs was cast onto the working electrode, and after drying, the probe was immobilized following a protocol reported in the literature[40].

The first step is the reduction of 100 µM probe in the presence of 10 mM triscarboxyethylphospine (TCEP) for 1 h. The resulting solution was then diluted to the chosen concentration (in the range of nanomolar) to be immobilized onto the AuNP-modified strip. A 20-µL drop of the probe was placed onto the working electrode area for 1 h at room temperature (to avoid solvent evaporation, the incubation was carried out in a humid chamber). After this step, the paper-based strip was gently rinsed with distilled water and incubated (in a humid chamber) with 2 mM mercaptohexanol to passivate the empty spaces onto the working electrode (1.5 h at room temperature). The strip was rinsed with distilled water and left in a humid chamber in a working buffer solution (50 mM phosphate buffer containing 150 mM NaCl (pH 7) until the experiment was carried out.

**Measurement of targets.** The measurements were carried out by using a single strip for each analysis. Briefly, the solution containing the target was drop casted on top of the working electrode of the strip, and the strip was incubated for 1 h at RT in a humid chamber. Subsequently, the strip has been gently washed with distilled water to remove target that was not attached to the immobilized probe on the strip. To perform the EIS measurement a PalmSens 4 potentiostat has been adopted, and the experiment has been carried by drop casting 100 µL of 1 mM $[Fe(CN)_6]^{3-/4-}$ on top of the strip. In case of blank measurements (in absence of the target) the strip has been firstly incubated with 50 mM phosphate buffer containing 150 mM NaCl (pH 7) for 1 h at RT in a

humid chamber, then gently washed with distilled water, and the EIS measurement was carried out in presence of $100\,\mu L$ of $1\,mM$ $[Fe(CN)_6]^{3-/4-}$ on top of the strip.

## Liposomes preparation, cellular and S-EV protocols

Liposomes preparation and characterization. Soy phosphatidylcholine (SPC) liposomes (10 mg/mL) were prepared by hydration of lipid film followed by extrusion. Briefly, the thin film was obtained by a rotary evaporator (Laborota 4010 digital, Heidolph, Schwabach, Germany) and then hydrated with $0.22\,\mu m$—filtered water for 30 min. The liposome suspension was extruded by a thermobarrel extruder system (Northern Lipids Inc., Vancouver, BC, Canada) forcing the suspension through polycarbonate membranes with decreasing pore sizes from 400 to 100 nm (Nucleopore Track Membrane 25 mm, Whatman, Brentford, UK). The size, polydispersity index (PI), zeta potential ($\zeta$) and liposomes concentration (n° of vesicles/ml of formulation) were determined by Zetasizer Ultra (Malvern Panalytical, Malvern, UK). Liposomes showed a mean diameter of 121.7 nm ± 2.0, a narrow size distribution as indicated by the PI (0.07 ± 0.04) and a zeta potential of 5.8 mV ± 1.2. Vesicles concentration was $1.04*10^{12} \pm 1.72*10^{11}$ of liposomes/mL.

## Cell culture

PC3 cell linewas purchased from American Type Culture Collection (Rockville, MD, USA). DU145R80 cells have been developed in Experimental Pharmacology Unit, from DU145 cells[33,34,41]. Both PC3 and DU145R80 cells were cultured in RPMI 1640 (Lonza) - containing 10% of heat-inactivated fetal bovine serum (FBS; Lonza), 10000 U/ml penicillin and 10 mg/mL streptomycin (Lonza), 20 mM Hepes (pH 7.4) and 4 mM L-glutamine in a humidified atmosphere composed of 95% air and 5% $CO_2$ at 37 °C. HCT116, A549, HEK293 and H460 cell lines were obtained from the American Type Culture Collections and grown in DMEM or RPMI medium (as recommended) supplemented with 10% heat-inactivated fetal bovine serum, penicillin (50 units/mL), streptomycin (500 µg/mL), and 4 mmol/L glutamine in a humidified atmosphere of 95% air and 5% $CO_2$ at 37°C. Cell lines were profiled by LGC Standards Cell Line Authentication service and regularly inspected for mycoplasma.

## S-EVs isolation

S-EVs were isolated from cell culture media of 24 h starved cell lines (50–100 $10^6$ cells). In details, cancer cells were grown in T175 flasks until they reached to 80% confluence, rinsed with PBS and cultured in serum-free media for 24 h. Supernatants were collected and centrifuged at $2000 \times g$ for 10 min at 4 °C to discard cellular debris. Next, media were centrifuged at $10,000 \times g$ for 30 min followed by filtration through 0.22-µm pore filters (Steriflip, Millipore) to remove large EVs. S-EVs were purified using the Total Exosome Isolation kit (Invitrogen, Carlsbad, CA ref. 4478359), according to the manufacturer's instructions. S-EVs pellets from each cell line were resuspended in 120 µL of 0.22 µm filtered PBS (TEM, device, TRPS) and RIPA BUFFER (wb) and stored at -80 °C until use (TRPS, TEM, WB).

## Tunable resistive pulse sensing measurements (TRPS)

S-EVs preparations (see above), suspended in 0.22 µm filtered PBS, were submitted to tunable resistive pulse sensing (TRPS) analysis using an Exoid instrument (IZON Science, Ltd, Christchurch, New Zealand) by a NP150 nanopore.

## Negative stain of isolated S-EVs and Transmission Electron Microscopy (TEM)

Negative stain treatment covers the sample with a layer of heavy metal salts so this contrast allows visualization of the sample surface. 5 µL of S-EVs were resuspended in PBS containing 4% Paraformaldehyde (1:1) to fix them. The sample was applied to formvar 100-mesh grids and incubated for 10 min. Grids were washed twice with filtered distilled water and stained using 1.5% UA in water for 10 min. After they were washed with water to remove the excess staining solution and grids were air-dried. Images were

acquired from grids using a FEI Tecnai 12 transmission electron microscope (FEI Company, Hillsboro, Oregon, USA) equipped with a Veleta CCD digital camera (Olympus Soft Imaging Solutions GmbH, Münster, Germany) and operating at 120 kV. Images were collected at magnifications of 21000X, 30000X and 68000X.

## Immuno-electron microscopy (IEM) of isolated S-EVs

For examination of immuno-gold labeled unfixed exosomes, rabbit anti-Integrin Alpha V + Beta 6 (1 µg/µL bs-5791R; Bioss Antibodies, Woburn, Massachusetts, USA) and Protein A gold-conjugate to 10 nm gold particles (PAG, purchased from Cell Microscopy Core Department of Cell Biology University Medical Center Utrecht, Utrecht, The Netherlands) were used. The first part of the protocol was identical to the fixation procedure for analysis of native exosomes. This was followed by an incubation step on drop of 0.2% Glycine/PBS 1 X for 2 min and then transfer to a drop of blocking with 1% Bovine Serum Albumin (BSA) for 20 min. Afterwards, grids were incubated for 2 h at room temperature on drop of the appropriately diluted primary antibody (anti-αvβ6 integrin at a 1:20 dilution in 1% BSA/PBS 1X), washed 2 times for 5 min in drops of 0.1% BSA/PBS 1X and binding of antibodies was detected with an additional incubation with Protein A conjugated with 10 nm colloidal gold (PAG, purchased from Cell Microscopy Core Department of Cell Biology University Medical Center Utrecht, Utrecht, The Netherlands) (dilution 1:50 in 1% BSA/PBS 1X) for 30 min at room temperature. Grids were then washed again 3 times for 5 min in filtered PBS 1X followed by post-fixation with 1% Glutaraldehyde, stained using 1.5% UA solution in water for 10 min. The excess UA solution on the grid was removed by contacting the grid edge with filter paper. Grids were washed 6 times for 1 min on a drop of filtered distilled water to remove the excess staining solution and air-dried.

Images were acquired with the same microscope FEI Tecnai 12 120 kV transmission electron microscope used for analysis of native exosomes.

## Immunoblot

25 µg of proteins lysates from cells grown in serum starved condition for 24 h and S-EVs (collected as described in "S-EVs isolation" section) were analysed by SDS-PAGE and Western Blot (WB) performed with antibodies: αvβ6 integrin (1:500 Bioss, bs-5791R), TSG101 (1:500, abcam cat. ab30871), CD81 (1:500, cell signaling #56039), β-actin (1:500, ab30871), γ-tubulin (1:500, sc-17787) Enhanced chemiluminescence (ECL) immunodetection reagents were from GE Healthcare. The chemiluminescent signal was detected with Image Quant LAS 500, and the intensity was measured by ImageQuantTL image software (GE Healthcare). Densitometric analysis was performed using NIH ImageJ software. Representative results from a single experiment of western blot were presented; additional experiments yielded similar results.

## Integrins profile

To profile integrins we took advantage of a colorimetric kit (α/β Integrin-mediated Cell Adhesion Array Combo Kit, Merck Millipore) following manufacturer's instructions. Briefly, following strips rehydration with 200 µL PBS/well, $1.5 \times 10^6$ EVs from each preparation were plated in Assay Buffer. After 2 h incubation at 37 °C, assay buffer was removed, S-EVs were washed and stained with a stain solution from the kit. Stain solution was removed and after several washes, stained S-EVs were allowed to dry and then red at 540–570 nm. The experiment was performed 2 times in duplicate.

## Reporting summary

Further information on research design is available in the Nature Portfolio Reporting Summary linked to this article.

## Data availability

The additional data supporting the findings of this study are available within the article and its Supplementary Information. The experimental datasets

used to produce Figures are available at the public repository https://zenodo.org/records/10682910.

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

## Acknowledgements

This research was funded by MUR-Ministero dell'Università e della Ricerca (Italian Ministry of University and Research), PRIN 2022 Prot. 2022YWZWB2, PON R&I 2014-2020-AIM (Attraction and International Mobility), project AIM1873131 - 2, linea 1. The research leading to these results received funding from AIRC under MFAG 2022 - ID. 27586 project – P.I. Cinti Stefano. The authors thank the EuroBioimaging facility located in IEOS-CNR (Castellino campus) for providing access to the imaging instruments used in this study. The Eurobioimaging facility is supported by the following grants: PON-IMPARA, SEELIFE and CIRO. The research was also supported by Italian Ministry of Health RC 2022-2024 (Linea 3/3) - PI Chiara Ciardiello.

## Author contributions

L.M., S.C. and C.C. designed the research; S.T., M.P., C.C., R.M., E.D.G., V.C., S.D.M., S.S. and A.R. performed the experiments; S.T., A.L., G.D., S.C., C.C., F.D.L., V.M.D.A., H.K., A.B. and L.M. analysed the data and wrote the manuscript.

## Competing interests

The authors declare no competing interests.
