## [Peer Review File · Communications Chemistry]

Reviewers' comments:

Reviewer #1 (Remarks to the Author):

The article entitled "Cancer Detection on Chip: A Nano-Engineered Electrochemical Device for Early Diagnosis of Integrin $\alpha\beta6$ Overexpressing Tumors" reported a nano-engineered paper-based electrochemical device for detecting cancer-derived small extracellular vesicles in fluids. The authors synthesized peptide probes and immobilized them onto the screen-printed electrode. The recognition of the target hindered the diffusion of ferro/ferricyanide towards the electrode, resulting in the increase of the EIS. In general, the manuscript is well-organized, So I recommend this manuscript for publication after some minor revisions towards the following points:

- (1) The EIS of different electrode may be different under the same conditions, thus, the repeatability and reproducibility of the electrodes (such as the RSD of EIS from different electrodes) was suggested to be added.
- (2) The resolution of some figures (Figure 2 and Figure 5 A) is low, making the figures not clear. Please revise it.
- (3) Figure 5B shows the TEM image of exosome, which is different from those on reported papers. You can easily find these images online.
- (4) Some closely related references should be cited. For example, the paper-based electrochemical biosensors (Biosens. Bioelectron. 2022, 199, 113906), electrochemical biosensors for exosome assay (Anal. Chem. 2021, 93, 11792, Anal. Chem., 2021, 93, 1709).

Reviewer #2 (Remarks to the Author):

The authors present a paper-based electrochemical device for the detection of small extracellular vesicles, achieving a limit of detection $< 10^3$ EV/mL and the possibility to detect the expression of $\alpha\beta6$ integrin receptor for cancer studies.

The reviewer considers the work interesting as well as the idea to develop an easy technique for early detection of cancer. Here the reviewer's suggestions:

- The core of the project is the capability to detect and discriminate cancer EV based on the detection of the integrin receptor concentration. The authors show enhanced values of the biomarker for all cells considered in the study (DU145R80 (R80), PC3, HCT116 and A549). Although the results are promising, the results are obtained only with positive controls. The reviewer considers very important for this study to consider different cell-lines associate to benign tumours or infections, in general not associated to malignant tumours, that could affect the biomarker levels. This is important to validate the system capability to discriminate not only cancer from no-cancer, but also from different forms of diseases.
- The state-of-the-art of configurations and techniques used for early detection of cancer, with a focus on paper-based microfluidics, should be improved, providing more information about sensitivity, resolution, specificity, multiplexing capability, in order to compare with those reported in this manuscript.
- The authors should comment and describe more in detail about the equivalent circuit model used to determine the sensor performance.

- What do the authors mean with “apparent binding constant”?
- Why is the quantification limit assumed as 3.3 LOD?
- The authors should replace figure 5 with a better-quality image. Furthermore, the term “particles diameters” related to the EV size is misleading and should be replaced with “EV size”.
- A description of the paper-based device should be discussed in the main manuscript and not only in the methods section.
- There are few typos in the manuscript, for example on page 2 line 64, “200um” should be replaced with 200nm.

The reviewer suggests to reply in detail to these comments and revise the manuscript accordingly before the acceptance.

Reviewer #3 (Remarks to the Author):

This paper reported the electrochemical device for detecting integrin $\alpha\text{v}\beta\text{6}$ which is overexpressed on cancer-derived small extracellular vesicles. The author developed a nano-engineered electrochemical paper-based device using electrochemical impedance spectroscopy. The author reported the probe synthesis which specifically captured the integrin $\alpha\text{v}\beta\text{6}$. However, there are a few concerns about analytical performance of this biosensor and other related issues. I would suggest further recommendation after the author discusses and provides additional information as outlined below:

1. I am afraid that the title “Cancer detection on chip” and “nano-engineered device” can be misleading. As from my understanding at first glance and after reviewing the manuscript, I expected some innovative and novelty on engineering a device. I found out that it was not a chip, it was paper-based strip. Moreover, I am not convinced that drop casting gold nanoparticles to modify carbon electrode and binding the probe with the thiolated linker were nano-engineered device and not sufficient to consider it as a novelty. I would suggest the amendment of the title to be more accurate and reflect the content in the manuscript.
2. Were gold nanoparticles saturated the working electrode? I am concerned that this factor can come to compromise the analytical performance. As the author used carbon conductive ink as a platform, casting gold nanoparticles may lead to batch-to-batch variation and difficult to control. Moreover, I suggested that the author should provide additional results on sensor surface characterisation. Please discuss.
3. From the previous point, I am firstly out of curiosity that why the author did not perform this electrochemical assay on a conventional gold screen-printed electrode instead of fabricating paper-based devices which were at risk to compromise the sensitivity and repeatability due to batch-to-batch variation. Moreover, as the author used a thiolated linker to bind the probe on a gold surface which is much easier to perform on conventional gold screen-printed electrode, why the author chose the electrochemical paper-based device as well as using drop-casting of gold nanoparticles. Please discuss.
4. The author should include the comparative comparison of the sensor performance with other biosensors. As for the best of my knowledge, I understand that there is no integrin $\alpha\text{v}\beta\text{6}$ detection as a

diagnostic tool reported elsewhere. Is there any similar type to be detected and reported? So that the author can make a relative comparison of the sensor performance and give an idea for the reader that this probe may help improve the detection performance. Please discuss.

5. From my understanding, the aim of this sensor is to detect integrin $\alpha\beta_6$ which is selected as a diagnostic tool in this study and reported the limit of detection and quantification in the level of 10^3 exosomes/mL, which is quite promising. However, the author already mentioned that testing with clinical samples was not included in this study. Is that possible for the author to provide the range of exosome concentration which is meaningful for clinical interpretation? I am concerned that if this range cannot distinguish two patient populations, then how achieving this LOD or LOQ is still meaningful. Please discuss.

Reviewer #1 (Remarks to the Author):

*The article entitled “Cancer Detection on Chip: A Nano-Engineered Electrochemical Device for Early Diagnosis of Integrin $\alpha\beta6$ Overexpressing Tumors” reported a nano-engineered paper-based electrochemical device for detecting cancer-derived small extracellular vesicles in fluids. The authors synthesized peptide probes and immobilized them onto the screen-printed electrode. The recognition of the target hindered the diffusion of ferro/ferricyanide towards the electrode, resulting in the increase of the EIS. In general, the manuscript is well-organized, So I recommend this manuscript for publication after some **minor revisions** towards the following points: (1) The EIS of different electrode may be different under the same conditions, thus, the repeatability and reproducibility of the electrodes (such as the RSD of EIS from different electrodes) was suggested to be added.*

We thank the Reviewer for the positive judice regarding the manuscript. According to the Reviewer’s suggestion, the RSD of EIS have been calculated from 8 different electrodes. In particular, 20 ng/mL of integrin has been measured with 8 different electrodes and a Rct of $39.5 \pm 4 \text{ k}\Omega$ has been obtained. The revised version of the manuscript has been improved with the following sentence:

“The repeatability of the platform was also evaluated by measuring 20 ng/mL of integrin with 8 different electrodes: a Rct of $39.5 \pm 4 \text{ k}\Omega$ has been obtained, corresponding to a relative standard deviation (RSD) of ca.11%, thus confirming the robustness of the paper-based chip manufacture.”

(2) The resolution of some figures (Figure 2 and Figure5 A) is low, making the figures not clear. Please revise it.

We thank the Referee for the suggestion, we have improved the resolution of both figures in the revised version of the manuscript.

(3) Figure 5B shows the TEM image of S-EVs, which is different from those on reported papers. You can easily find these images online.

We thank the Reviewer for the suggestion and replaced the old picture with new one, representative of other fields. These pictures are similar to those reported in some published papers (Mol Cell Proteomics.2012 Oct;11(10):863-85. doi: 10.1074/mcp.M111.014845; Biosens Bioelectron.2017 Aug 15:94:400-407.doi: 10.1016/j.bios.2017.03.036; Pract Lab Med.2018 Apr 22:12:e00099. doi: 10.1016/j.plabm.2018.e00099). In this revised version of the manuscript, we have also added more TEM images, representative of our population of S-EVs.

(4) Some closely related references should be cited. For example, the paper-based electrochemical biosensors (Biosens. Bioelectron. 2022, 199, 113906), electrochemical biosensors for S-EVs assay (Anal. Chem. 2021, 93, 11792, Anal. Chem., 2021, 93, 1709).

As per referee suggestion, closely related references to both, paper-based and electrochemical biosensors for S-EVs assay have been inserted in the manuscript. We thank the referee for the advice.

Reviewer #2 (Remarks to the Author):

The authors present a paper-based electrochemical device for the detection of small extracellular vesicles, achieving a limit of detection $< 10^3$ EV/mL and the possibility to detect the expression of $\alpha\beta6$ integrin receptor for cancer studies. The Reviewer considers the work interesting as well as the idea to develop an easy technique for early detection of cancer. Here the Reviewer's suggestions: - The core of the project is the capability to detect and discriminate cancer EV based on the detection of the integrin receptor concentration. The authors show enhanced values of the biomarker for all cells considered in the study (DU145R80 (R80), PC3, HCT116 and A549).

Although the results are promising, the results are obtained only with positive controls. The reviewer considers very important for this study to consider different cell-lines associate to benign tumours or infections, in general not associated to malignant tumours, that could affect the biomarker levels. This is important to validate the system's capability to discriminate not only cancer from no-cancer, but also from different forms of diseases.

The authors deeply thank the Referee for the observation, which gives us the opportunity to clarify some issues that were not enough explained in the original paper and to add new cell lines to further challenge the biosensor.

a) Concerning the utilization of S-EVs from non-cancerous diseases “so to validate the system's capability to discriminate not only cancer from no-cancer, but also from different forms of diseases”, we highlight that, as far as our current understanding extends, the application of S-EVs for diseases such as neurodegenerative, cardiovascular, or infectious conditions is still in its early stages. Even less information is known about the role of $\alpha\beta6$ in these diseases. To the best of our knowledge, the singular exception, aside from cancers, where $\alpha\beta6$ is overexpressed, is idiopathic pulmonary fibrosis—a rare condition affecting 0.09-1.30 individuals per 10,000.

b) In reference to the utilization of solely $\alpha\beta6$ -positive cells for the validation of the biosensor, we extend our sincere appreciation to the Reviewer for bringing attention to this crucial aspect. We acknowledge that our previous communication on this intricate matter may not have been sufficiently clear. To elaborate, for testing the biosensor designed for $\alpha\beta6$ -expressing tumors, we initially selected two well-established $\alpha\beta6$ -expressing cancer lines, namely R80 and PC3 prostate cancer cells (as detailed in J.Biol. Chem. 2015, 290, 4545–4551). Additionally, we incorporated two cell lines with very low $\alpha\beta6$ expression levels—specifically, A549 lung cancer cells and colon cancer HCT116 cells—both characterized in the literature as $\alpha\beta6$ -negative cells due to their minimal integrin expression (Nat. Commun. 2021, 12, 5209). Furthermore, as an unequivocal negative control, we included a liposome of identical size and lipidic composition to that of S-EVs.

However, it is noteworthy to mention that despite the low membrane expression of $\alpha\beta6$ on lung cancer A549 and colon cancer HCT116 cells, their released S-EVs exhibited an enrichment of this integrin. While the proteome of S-EVs is known to closely mirror that of the donor cells, studies have

demonstrated a specific enrichment of heparin-binding receptors, phospholipid-binding proteins, and extracellular matrix-binding receptors such as integrins (Proteomics 2019, 19, e1800167, doi: 10.1002/pmic.201800167). Consequently, even when utilizing a cell line classified as " $\alpha\beta6$ negative," the S-EVs released by these cells still demonstrated $\alpha\beta6$ positivity, albeit to a lesser extent when compared to S-EVs from $\alpha\beta6$ -expressing cells. This underscores the fact that, although S-EVs are generally enriched with integrins, they effectively reflect the quantity of integrins, including $\alpha\beta6$, present in the cells of origin. It is essential to note that this intricacy has been elucidated and emphasized in the revised version of the paper.

Given these considerations, and in response to the valuable suggestion from the referee, we extended our investigation by incorporating two additional cell lines: a non-cancerous cell line (HEK293, derived from human embryo's kidney) and a second non-small cell lung cancer (NSCLC)-derived cell line (H460). These cell lines are documented in the literature to exhibit low expression of $\alpha\beta6$ integrin (see PMID: 21752268 for HEK293 and PMID: 29116098 for H460). Consistent with our previous findings for A549 and HCT116, the S-EVs derived from these new cell lines demonstrated $\alpha\beta6$ positivity, with those from the healthy cells featuring the lowest amount of $\alpha\beta6$ among all tested.

In light of these results, A549, HCT116, HEK293, and H460 are all considered negative controls, while R80 and PC3 serve as positive controls. The biosensor demonstrated its ability to effectively distinguish between $\alpha\beta6$ -expressing and $\alpha\beta6$ -low expressing cells, irrespective of their cellular origin. In this context, it is conceivable that above a certain threshold of $\alpha\beta6$ detection (defined herein for S-EVs from cell cultures, but subject to potential variations in diverse human biofluids, warranting further dedicated studies), a cancer diagnosis becomes plausible, while below this threshold, it may not be feasible—similar to the interpretation of many known cancer biomarkers.

To illustrate, established cancer biomarkers like CA125 and CA199 are not exclusively absent in healthy cells or specific to tumor cells. Instead, their diagnostic value lies in concentration thresholds. For instance, a CA125 concentration greater than 35 U/ml in serum suggests a potential malignant tumor, despite being elevated in other cancers. Similarly, CA199, a sensitive marker for pancreatic cancer, is indicative of potential malignant tumors when its serum concentration exceeds 22 U/ml, even though it is also elevated in other tumors such as gastric, lung, and colorectal cancers. Following this logic, we do not expect that a promising cancer biomarker like $\alpha\beta6$ integrin would be entirely absent in healthy cells or exclusively expressed by tumor cells. Instead, its significance is likely tied to the rise in concentration in S-EVs, providing an indicative measure of a malignancy.

In summary, our biosensor has been developed to detect $\alpha\beta6$ -expressing tumors (even though only prostate cancer was here evaluated as proof of concept). Notably, $\alpha\beta6$ is widely expressed in various cancers, including but not limited to those affecting the pancreas, breast, prostate, lung, oral, skin, colon, and stomach. As such, the potential utility of our biosensor spans across a broad spectrum of cancers.

To enhance the specificity and reliability of biosensor-based cancer diagnosis, the incorporation of one or two specific recognition probes for additional cancer markers can be considered. The choice of these

probes would be tailored to the specific tumor one aims to screen, thereby augmenting the biosensor's versatility in different cancer contexts.

We express our sincere gratitude to the referee for their insightful comments, which have significantly contributed to clarifying and refining the explanation of this intricate subject.

- The state-of-the-art configurations and techniques used for early detection of cancer, with a focus on paper-based microfluidics, should be improved, providing more information about sensitivity, resolution, specificity, multiplexing capability, in order to compare with those reported in this manuscript. The authors should comment and describe more in detail about the equivalent circuit model used to determine the sensor performance. - What do the authors mean by “apparent binding constant”?
- Why is the quantification limit assumed as 3.3 LOD?

As for Referee's suggestion, the introduction has been improved describing the advantages and limitations of current colorimetric and electrochemical portable devices to be applied in liquid biopsy. We have better discussed the comparison of the developed platform with others already existing, and we have added a new Table of comparison in the Supporting Information file (Table S6 in SI). We thank the Referee for the useful suggestion.

- The authors should comment and describe more in detail about the equivalent circuit model used to determine the sensor performance.

The Authors thank this Referee for the suggestion, that surely strengthen the significance of the manuscript. Thus, as suggested, a novel description has been added in the revised version of the manuscript, as follows: “The impedance spectra obtained are composed of a semicircle at high frequencies, followed by a 45° straight line at low frequencies. The equivalent circuit that describes this trend is a typical Randles circuit, composed by the resistance of the solution R_s in series with the capacitance of the electric double layer CPE and the charge transfer resistance R_{ct} (in series with an open Warburg element Z_w , that describes the mass transport), respectively in parallel. The double-layer electrical capacitance was fitted with a CPE, since the electrode surface is not completely homogeneous. R_{ct} is the only parameter proportional to the analyte concentration and was used to produce the calibration curves, while the other parameters remain constant, as expected.”

- What do the authors mean with “apparent binding constant”?

The electrochemical architecture that has been applied in this study is based on a covalently immobilized probe which is able to recognize integrin in solutions. However, being the probes immobilized onto a surface, both the detection limit and specificity of these sensors usually are not dependent on the “true” probe-target affinity but depend on probe concentration; consequently, the response of the sensor platform depends on the density with which the probes are packed on the surface of the sensor, thus leading to an estimation of an “apparent” or “observed” affinity, as reported in a recent paper published in the field of analytical chemistry (<https://doi.org/10.1021/ac4012123>). This sentence has been added

to the revised version of the manuscript, and the references have been also updated, as follows: “It should be clarified that, being the probes immobilized onto a surface, both the detection limit and specificity of these sensors usually are not dependent on the “true” probe-target affinity but depend on probe concentration; consequently, the response of the sensor platform depends on the density with which the probes are packed on the surface of the sensor, thus leading to an estimation of an “apparent” or “observed” affinity, as previously reported.”

- *Why is the quantification limit assumed as 3.3 LOD?*

The Authors thank for the meaningful question. The LOD has been calculated based on the standard deviation of the response of the curve (determined on the standard deviation of blank measurements) and the slope of the calibration curve, according to the formula, $LOD = 3.3(SD/slope)$. The LOQ has been calculated equal to $10(SD/slope)$. All the information and the resulting values have been revised accordingly.

- The authors should replace figure 5 with a better-quality image. Furthermore, the term “particles diameters” related to the EV size is misleading and should be replaced with “EV size”.

The Authors thank the Referee for the suggestions. Figure 5 has been replaced with a better-quality image and the term “particles diameters” has been replaced with “EV size”.

- *A description of the paper-based device should be discussed in the main manuscript and not only in the methods section.*

Following the Referee’s suggestion, a description of the paper-based device the revised version of the manuscript has been added to the main text.

- *There are few typos in the manuscript, for example on page 2 line 64, “200um” should be replaced with 200nm.*

We thank the Referee for highlighting this issue. Accordingly, the typo on page 2 has been corrected and the whole manuscript carefully revised.

Reviewer #3 (Remarks to the Author):

This paper reported the electrochemical device for detecting integrin $\alpha v\beta 6$ which is overexpressed on cancer-derived small extracellular vesicles. The author developed a nano-engineered electrochemical paper-based device using electrochemical impedance spectroscopy. The author reported the probe

synthesis which specifically captured the integrin $\alpha\beta6$. However, there are a few concerns about analytical performance of this biosensor and other related issues. I would suggest further recommendation after the author discusses and provides additional information as outlined below:

1. I am afraid that the title “Cancer detection on chip” and “nano-engineered device” can be misleading. As from my understanding at first glance and after reviewing the manuscript, I expected some innovative and novelty on engineering a device. I found out that it was not a chip, it was paper-based strip. Moreover, I am not convinced that drop casting gold nanoparticles to modify carbon electrode and binding the probe with the thiolated linker were nano-engineered device and not sufficient to consider it as a novelty. I would suggest the amendment of the title to be more accurate and reflect the content in the manuscript.

We thank the Referee for the suggestion. Accordingly, the title was modified in “Paper-based Electrochemical Device for Early Detection of Integrin $\alpha\beta6$ expressing Tumors”

2. Were gold nanoparticles saturated the working electrode? I am concerned that this factor can come to compromise the analytical performance. As the author used carbon conductive ink as a platform, casting gold nanoparticles may lead to batch-to-batch variation and difficult to control. Moreover, I suggested that the author should provide additional results on sensor surface characterisation. Please discuss.

We thank the Referee for the suggestion. To evaluate the batch-to-batch variation, three different batches of synthesized gold nanoparticles were used to modify electrodes, and by performing a cyclic voltammetry in presence of 0.1 M sulfuric acid. The electrodes displayed similar peaks attributable to the presence of similar amount of gold nanoparticles on the working surface, as reported in Supporting Information file, as Figure S5.

3. From the previous point, I am firstly out of curiosity that why the author did not perform this electrochemical assay on a conventional gold screen-printed electrode instead of fabricating paper-based devices which were at risk to compromise the sensitivity and repeatability due to batch-to-batch variation. Moreover, as the author used a thiolated linker to bind the probe on a gold surface which is much easier to perform on conventional gold screen-printed electrode, why the author chose the electrochemical paper-based device as well as using drop-casting of gold nanoparticles. Please discuss.

The Authors thank the Referee for the observation, which gives us the opportunity to highlight a further advantage of our novel platform: its cheapness. In fact, the decision to use paper-based electrodes modified with gold nanoparticles was due to the cost of the platforms. For instance, commercial gold electrodes (from Metrohm, product code 220AT) are characterized by a cost of ca. 3 Euros/each. As reported in our previous work, our electrodes, obtained onto office paper are characterized by a cost of ca. 0.02 Euro/each (<http://dx.doi.org/10.1016/j.snb.2017.07.161>). A sentence describing this feature has been added to the revised version of the manuscript.

4. The author should include the comparative comparison of the sensor performance with other biosensors. As for the best of my knowledge, I understand that there is no integrin $\alpha\beta6$ detection as a diagnostic tool reported elsewhere. Is there any similar type to be detected and reported? So that the author can make a relative comparison of the sensor performance and give an idea for the reader that this probe may help improve the detection performance. Please discuss.

We thank the Reviewer for the suggestion, that surely allows us to improve the quality of the manuscript. We have better discussed the comparison of the developed platform with others already existing, and we have added a new Table of comparison in the Supporting Information file (Table S6 in SI). The text was improved as follows:

“As far as we know, the development of a portable, electrochemical device for the recognition of $\alpha\beta6$ appears as novel in literature. As shown in Table S6 (see SI), the detection limit obtained in this work satisfactorily compares with other sensing approaches applied towards integrin receptors, and the electrochemical methods reported here is more user friendly of the largely reported SERS and FRET-based architectures.”

5. From my understanding, the aim of this sensor is to detect integrin $\alpha\beta6$ which is selected as a diagnostic tool in this study and reported the limit of detection and quantification in the level of 10^3 S-EVs/mL, which is quite promising. However, the author already mentioned that testing with clinical samples was not included in this study. Is that possible for the author to provide the range of S-EVs concentration which is meaningful for clinical interpretation? I am concerned that if this range cannot distinguish two patient populations, then how achieving this LOD or LOQ is still meaningful. Please discuss.

The Authors express sincere gratitude to the Reviewer for bringing up this crucial point for discussion. To the best of our knowledge, the quantity of extracellular vesicles (EVs) in human blood is inherently variable and subject to influence from diverse factors, including health conditions. Furthermore, accurately quantifying the specific number of exosomes in a biological fluid remains a formidable challenge, primarily due to the absence of standardized methods for isolation and quantification. Despite these challenges, existing studies have indicated that the quantity of exosomes in healthy plasma ranges from 10^8 to 10^{10} particles/mL, with amounts reaching up to 10^{14} particles/mL in the plasma of cancer patients (PMID: 26044649; PMID: 22691960). Therefore, the limit of detection and quantification at 10^3 S-EVs/mL of our device appears to be suitable for a clinical setting, even though the precise quantity of $\alpha\beta6$ -expressing S-EVs in cancer patients remains unknown. Recognizing the pivotal significance of this issue in advancing our biosensor, we are acutely aware of the need to address it comprehensively. Consequently, we have outlined plans to undertake a dedicated study involving biological samples from a substantial cohort of $\alpha\beta6$ -expressing cancer-bearing patients to shed light on this aspect.

Yours sincerely,

Luciana Marinelli, PhD

REVIEWERS' COMMENTS:

Reviewer #1 (Remarks to the Author):

In this revised manuscript, the authors have appropriately addressed the questions raised by the reviewers. Hence, I suggest its acceptance for publication in this journal.

Reviewer #2 (Remarks to the Author):

The authors have replied to the reviewer's comments point-by-point in an exhaustive way, improving the quality of the work.

The reviewer suggests for the acceptance of the manuscript.

Reviewer #3 (Remarks to the Author):

I am satisfied with the response and the current form of manuscript. I recommend to accept the publication.

Minor points:

Figure 4, it would be good if the author changed the scale of 2 graphs to be the same and included statistical analysis to show it is statistically significant different.

REPLY to REFEREE 3: "Figure 4, it would be good if the author changed the scale of 2 graphs to be the same and included statistical analysis to show it is statistically significant different" We really thank the Referee for its suggestion. As requested, we modified the Figure 4 using the same scale and we added the % value on top the avb6 Integrin histograms that is relative to the relative standard deviation obtained on 8 different devices, as also reported in the manuscript.